# Extensive screening reveals previously undiscovered aminoglycoside resistance genes in human pathogens

David Lund [1,2], Roelof Dirk Coertze[2,3], Marcos Parras-Moltó[1,2], Fanny Berglund [2,3], Carl-Fredrik Flach [2,3], Anna Johnning [1,2,4], D. G. Joakim Larsson [2,3] & Erik Kristiansson [1,2 ✉]

Antibiotic resistance is a growing threat to human health, caused in part by pathogens accumulating antibiotic resistance genes (ARGs) through horizontal gene transfer. New ARGs are typically not recognized until they have become widely disseminated, which limits our ability to reduce their spread. In this study, we use large-scale computational screening of bacterial genomes to identify previously undiscovered mobile ARGs in pathogens. From ~1 million genomes, we predict 1,071,815 genes encoding 34,053 unique aminoglycoside-modifying enzymes (AMEs). These cluster into 7,612 families (<70% amino acid identity) of which 88 are previously described. Fifty new AME families are associated with mobile genetic elements and pathogenic hosts. From these, 24 of 28 experimentally tested AMEs confer resistance to aminoglycoside(s) in *Escherichia coli*, with 17 providing resistance above clinical breakpoints. This study greatly expands the range of clinically relevant aminoglycoside resistance determinants and demonstrates that computational methods enable early discovery of potentially emerging ARGs.

[1] Department of Mathematical Sciences, Chalmers University of Technology and University of Gothenburg, Gothenburg, Sweden. [2] Centre for Antibiotic Resistance Research (CARe), University of Gothenburg, Gothenburg, Sweden. [3] Department of Infectious Diseases, Institute of Biomedicine, Sahlgrenska Academy, University of Gothenburg, Gothenburg, Sweden. [4] Department of Systems and Data Analysis, Fraunhofer-Chalmers Centre, Gothenburg, Sweden. ✉email: erik.kristiansson@chalmers.se

Antibiotic resistance keeps spreading among pathogens, threatening to irrevocably lessen the usefulness of antibiotics in treating and preventing bacterial infections[1]. Resistance commonly arises when microorganisms acquire mobile antibiotic resistance genes (ARGs) through horizontal gene transfer[2]. This process is usually facilitated by mobile genetic elements (MGEs), such as conjugative plasmids and insertion sequences, which can allow ARGs to rapidly disseminate between bacterial cells in communities under sufficient selection pressures[3,4]. Pathogens are constantly becoming more resistant through the accumulation of new, often more efficient, ARGs[5,6]. However, the lack of knowledge about the evolutionary processes behind this ongoing gene transfer makes effective preventative measures hard to implement.

Aminoglycosides constitute an important class of antibiotics for which clinical resistance is increasing[7,8]. These compounds have a long history of clinical use, primarily as a treatment for infections by Gram-negative bacteria (e.g. Enterobacteriaceae), but also as a second-line treatment against specific Gram-positive pathogens (e.g. multidrug-resistant *Mycobacterium tuberculosis*)[9,10]. Resistance to aminoglycosides is associated with several mechanisms, where drug inactivation by aminoglycoside-modifying enzymes (AMEs) is most common in clinical settings[11,12]. Among the AME mechanisms, the most abundant are the aminoglycoside acetyltransferases (AACs) and the aminoglycoside phosphotransferases (APHs)[13,14]. The AACs act by acetylating aminoglycosides at position 1 (AAC(1)), 2' (AAC(2')), 3 (AAC(3)), or 6' (AAC(6')). The majority of these enzymes belong to the large GCN5-related *N*-acetyltransferase (GNAT) family of proteins[15,16], except most AAC(3) enzymes[17] and, potentially, AAC(1) enzymes for which no sequences have been made publicly available[13]. The APHs, on the other hand, use phosphorylation at position 2" (APH(2")), 4 (APH(4)), 3' (APH(3')), 3" (APH(3")), 6 (APH(6)), 7" (APH(7")), or 9 (APH(9)) to inactivate aminoglycosides. The origins of APHs are not clear, but they have been hypothesized to share an evolutionary history with Mph macrolide phosphotransferases and eukaryotic protein kinases (e.g. cAMP-dependent protein kinase cAMP) due to their structural similarity[18,19]. Among the AMEs, AACs have the highest known diversity, with 86 gene sequences present in ResFinder[20], compared to 39 gene sequences of APHs reported to date. However, the full diversity of these enzymes remains unknown.

Bacterial communities present in humans, animals, and external environments are known to maintain a large diversity of ARGs. This includes genes encoding AMEs and constitutes a reservoir from which genes can be recruited into pathogens[21–24]. The recent origins of most ARGs are still unknown, which suggests that they were likely mobilized from species that are not yet well-represented in current sequence repositories[25]. Consequently, new emerging ARGs are often identified after they become widespread in pathogens and thus constitute a substantial clinical problem. For example, the beta-lactamase NDM-1 was originally discovered in a *Klebsiella pneumoniae* infection acquired from India in 2008[26], but already by 2009, the gene was frequently encountered in clinics in India, Pakistan, Bangladesh, and the UK[27]. This rapid emergence suggests that NDM-1 was commonly carried by pathogens before its discovery. Similarly, the plasmid-encoded colistin resistance determinant MCR-1, which was first observed in commensal and clinical isolates from China in 2015, was present in at least 16 countries across two continents at the time of its discovery[28,29]. The inability to stop emerging resistance genes lead not only to increased morbidity and mortality of patients but also to growing healthcare costs[30]. Indeed, the costs resulting from a single outbreak of NDM-carrying bacteria at a hospital ward have been estimated to be in the order of 1 million US dollars[31]. To protect the efficacy of existing and future antibiotics, emerging ARGs should ideally be identified at an early stage. This would enable gene-specific diagnostics and facilitate the implementation of countermeasures, such as targeted infection control and surveillance, to limit further dissemination.

A fundamental challenge in the detection of emerging ARGs is their general absence from the sequence databases, making it difficult to identify them using standard methods. New ARGs can be detected using functional metagenomics[32], but this technique is costly and has limited throughput, and is therefore hard to implement at a large scale. Alternatively, a wide range of computational methods has been developed for the prediction of uncharacterized ARGs from sequencing data[33]. These methods typically leverage the current resistance gene databases, like ResFinder[20] and CARD[34], and use models to predict genes based on their sequence or structural features. Examples include fARGene, which uses gene-specific hidden Markov models (HMMs) to identify homologous ARGs with low sequence similarity;[35] deepARG, which applies deep learning algorithms to discriminate between new ARGs and other genes;[36] and PCM[37] and ARGGNN[38], both of which use machine learning to identify new ARGs based on features from their protein structure. Experimental validation has shown that the general accuracy for computational prediction of ARGs is as high as >80%[35], making these methods a reliable option for detecting new resistance genes.

In this study, we demonstrate how computational prediction can be used to identify previously undiscovered aminoglycoside resistance genes. By analyzing ~1 million bacterial genomes, we predicted 1,071,815 genes encoding 34,053 unique AMEs, divided into 7,612 AME families (<70% amino acid identity). Among these, 50 new AME families contained genes that were co-localized with MGEs and present in human pathogens. Experimental validation of genes from 28 of these families showed that 24 induced a resistance phenotype in *Escherichia coli*, of which 17 conferred resistance above clinical breakpoints and/or epidemiological cut-off values (ECOFFs). Our results greatly expand the number of known AMEs and provide a unique view of the uncharacterized but clinically relevant aminoglycoside resistance determinants. We conclude that large-scale computational screening of bacterial genomes constitutes an efficient way to identify new resistance genes before they become widespread in pathogens.

## Results

**Development of gene models for the prediction of new aminoglycoside-modifying enzymes.** New genes encoding AMEs were predicted in bacterial genomes using fARGene, a software that utilizes optimized HMMs to identify ARGs in sequence data[35]. Nine HMMs (A-I) were created, representing genes from two major mechanisms of AMEs: six models (A-F) for AACs and three models (G-I) for APHs, based on phylogeny (Supplementary Figs. 1, 2, 3). The sensitivity of the models was optimized based on experimentally validated protein sequences unique to each model (Supplementary Data 1). To ensure high specificity, the models were also optimized against negative datasets consisting of genes with an evolutionarily close relationship to AACs or APHs that have not been reported to confer aminoglycoside resistance (Supplementary Data 1; see Methods for full details). After cross-validation, the models displayed high sensitivity and specificity, with all but one model showing a sensitivity of 1.0 (0.9375 for Model E [AAC(6')-I]) while the specificity of all models was 1.0 at the optimized threshold score (Supplementary Fig. 4, Supplementary Table 1).

**Table 1 Comparison between aminoglycoside resistance genes predicted in pathogenic species and the total collection of predicted genes.**

| Model | Representative phenotype(s) | Pathogens | | All species | |
|---|---|---|---|---|---|
| | | Predicted AMEs [unique proteins] | AME Families[a] | Predicted AMEs [unique proteins] | AME Families[a] |
| A | AAC(2')-I | 296 [41] | 6/4 | 4,715 [1,645] | 341/7 |
| B | AAC(3)-I | 1,755 [36] | 1/4 | 3,061 [832] | 360/8 |
| C | AAC(3)-II, III, VI, VII, VIII, IX | 102,582 [1,028] | 18/6 | 116,274 [6,961] | 1,469/13 |
| D | AAC(6')-I,II | 19,015 [526] | 8/8 | 24,754 [2,248] | 513/9 |
| E | AAC(6')-I | 436,848 [998] | 35/8 | 446,428 [5,021] | 1,185/9 |
| F | AAC(6')-I | 20,899 [110] | 4/6 | 23,087 [886] | 349/8 |
| G | APH(2'')-I | 3,403 [305] | 9/6 | 5,172 [1,251] | 386/6 |
| H | APH(3')-I, II, III, IV, V, VI, VII, VIII, IX, XV + APH(3'')-I | 231,817 [1,745] | 30/14 | 248,178 [6,005] | 1,146/22 |
| I | APH(6)-I + APH(3')-II | 180,758 [1,286] | 33/4 | 200,146 [9,204] | 1,776/6 |
| Total | | 997,373 [6,075] | 144/60 | 1,071,815 [34,053] | 7,525/88 |

[a] = New/known.

**Fig. 1 The number and distribution of the predicted AME families, divided into known and new AMEs. a** AME families carried by pathogens. **b** AME families carried by all species. **c** Mobile AME families carried by pathogens. **d** Mobile AME families carried by all species.

**Identification of new aminoglycoside-modifying enzymes in pathogens.** Next, we used the optimized models and screened 990,306 bacterial genomes downloaded from NCBI Assembly[39] for genes encoding AMEs. This yielded a total of 1,071,815 predicted genes – 153,317 of which were previously unknown (Supplementary Data 2) – encoding 34,053 unique protein sequences (after clustering at 100% amino acid identity; Table 1) representing both known and new AMEs. The predicted AMEs clustered into 7,613 AME families (< 70% amino acid identity), of which 4,271 represented AACs and 3,342 families represented APHs. Among the analyzed genomes, 280,372 (28.3%) encoded a single AME, while 261,295 (26.4%) encoded multiple variants. As many as 997,373 AMEs (93%) were predicted in pathogenic species, but as these only corresponded to 6,075 unique proteins (17.8%) across 204 AME families (2.7%) they merely represented a fraction of the genetic diversity predicted across all species. However, considering that only 112 (0.6%) of the species carrying AMEs were classified as pathogenic, the amount of new AMEs predicted in pathogenic species was considerable (Fig. 1a). This was especially true for genes predicted by model C [AAC(3)-II, III, VI, VII, VIII, IX], E [AAC(6')-I], and I [APH(6)-I + APH(3')-II], where 75%, 81%, and 89% of the pathogen-associated predicted AME families were new, respectively. However, when compared to non-pathogenic species, pathogens generally carried a lower proportion of new AMEs (Fig. 1b).

**New AMEs in pathogens show high potential for mobility.** Phylogenetic analysis of the AMEs predicted by each model was used to analyze their evolutionary history. Clear deviations between the gene and host phylogenies were observed for all models, indicating multiple horizontal gene transfer events (Supplementary Fig. 5). To further investigate the potential mobility of the new AMEs found in pathogens, we screened their genetic context for MGEs. We identified 104 pathogen-associated AME families, of which 50 (48%) were not present in current ARG repositories, whose members co-localized with conjugative elements, insertion sequences, integrons, and/or other known mobile ARGs (Figs. 1c and 2). Most new mobile pathogen-associated AME families (36) were found in proteobacterial pathogens (Pseudomonas aeruginosa being the most common). However, pathogenic species from Firmicutes (e.g., Streptococcus sp.) and Bacteroidetes (Elizabethkingia anophelis) were also present among the carriers. Notably, several AME families (e.g., C24, H510, see Supplementary Data 3), included genes present in multiple phyla, suggesting that these genes have at some point undergone inter-phyla horizontal gene transfer.

The new mobile pathogen-associated AME families were generally associated with specific types of MGEs, with only 10 families (20%) co-localizing with elements from multiple categories. The most common indication of MGEs were genes involved in conjugation – such as relaxases and/or mating pair formation genes – which were co-localized with AMEs from 34 new pathogen-associated families (68%; Fig. 2a). These were primarily mating pair formation genes from the F, G, or FATA classes and the $MOB_{P1}$ relaxase, which are associated with broad host ranges[40]. By contrast, the known pathogen-associated AMEs were associated with a wider variety of MGEs, with 19 AME families (32%) co-localizing with elements from all included

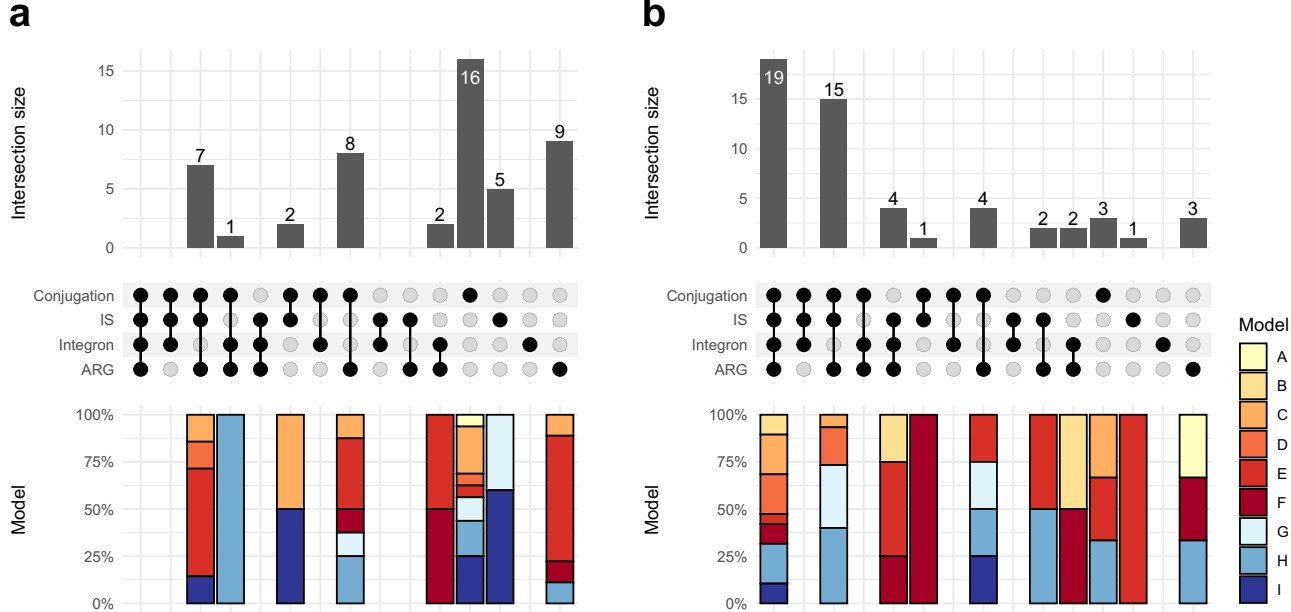

**Fig. 2 The number of AME families carried by pathogenic species that were associated with different combinations of genes relating to mobile genetic elements (MGEs), including conjugation systems, insertion sequences (IS), integrons, and/or other known mobile antibiotic resistance genes (ARGs).** The bars at the bottom indicate the distribution of genes predicted by each of the nine models within each category. **a** Families representing new AMEs and **b** Families representing known AMEs.

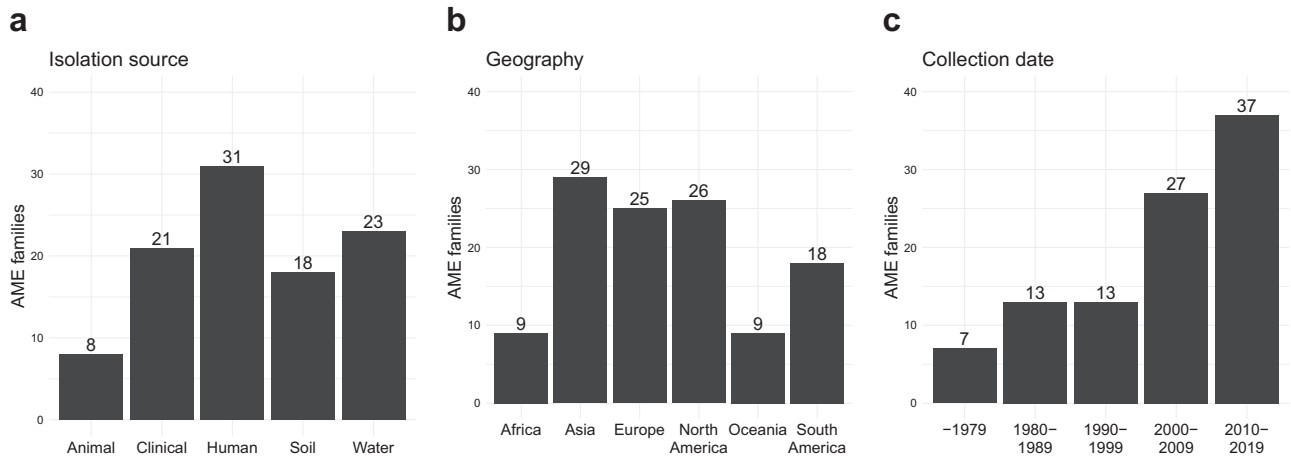

**Fig. 3 Metadata about the isolates (**$n = 92,056$**) carrying new AMEs that were associated with mobile genetic elements in pathogens (representing 50 AME families).** The panels show the number of AME families divided based on their hosts' isolation **a** environment (data available for 14,747 isolates (16%) representing 39 AME families (78%)), **b** continent (data available for 10,631 isolates (12%) representing 41 AME families (82%)), and **c** collection date (data available for 10,260 isolates (11%) representing 41 AME families (82%)). Note that a family can contain gene variants carried by multiple hosts from different sources.

categories. Conjugation was indicated to also be the main mechanism by which known AMEs are transferred, as genes from 42 known pathogen-associated families (78%) appeared close to conjugation genes (Fig. 2b). Furthermore, genes from 27 of the new pathogen-associated AME families (54%) co-localized with other mobile resistance genes, suggesting that co-selection can play a part in their dissemination. These co-localized ARGs included genes conferring resistance to beta-lactams, tetracyclines, macrolide, quinolones, and sulfonamides, as well as other aminoglycoside resistance genes (Supplementary Data 3).

Next, we analyzed the source environment, geographical location, and collection date of the isolates containing the new mobile pathogen-associated AMEs. From this, we found that 21 families (42%) contained at least one gene identified in a clinical isolate (Fig. 3a), and that hosts of the new mobile pathogen-associated AME families were found globally (Fig. 3b). Interestingly, genes from seven families were found in genomes isolated before 1979, suggesting that they have been present in pathogens for a long time (Fig. 3c).

**Most of the new AMEs induce a resistance phenotype**. We assessed the functionality of the predicted AMEs by expressing representative genes from 28 new mobile pathogen-associated AME families in *E. coli*. The resulting phenotypes were evaluated through disk-diffusion tests with seven different aminoglycosides. Genes were selected for evaluation based on their potential risk to human health through the combination of host species pathogenicity, co-localized MGE-associated genes, and other ARGs

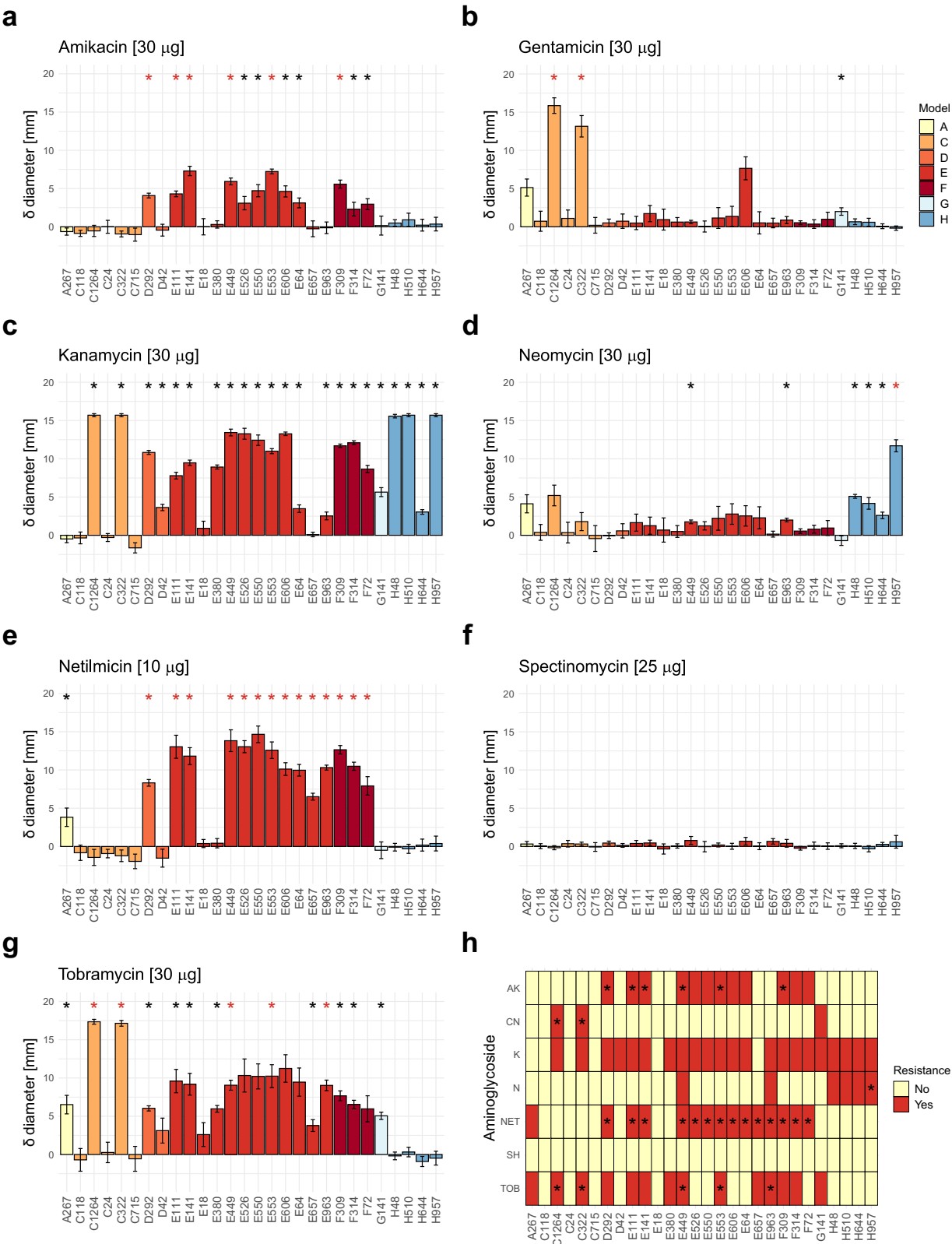

(Supplementary Data 3), as well as the likelihood of producing a measurable resistance phenotype under the tested conditions (assuming a similar phenotype to the closest known homolog). Of the 28 tested ARGs, 24 (86%) conferred significantly increased resistance (i.e., decreased susceptibility compared to control with $p$-value < 0.01, exact $p$-values for each test are shown in Supplementary Data 4) to at least one aminoglycoside, while 23 (82%)

conferred resistance to multiple aminoglycosides in *E. coli* (Fig. 4). While several of the tested genes provided significantly increased resistance to amikacin, gentamicin, kanamycin, neomycin, netilmicin, and tobramycin (12, 3, 22, 6, 12, and 11 genes, respectively), none induced resistance to spectinomycin. When comparing the inhibition zone diameters to EUCAST's clinical breakpoints and ECOFFs[41], 17 of the tested genes (61%)

**Fig. 4 Results from disk diffusion tests using *E. coli* and 28 selected new AMEs.** Panels **a–g** show the mean inhibition zone diameter difference [mm] between clones carrying new AMEs ($n = 3$ for each gene and antibiotic) and susceptible controls ($n = 6$ for each antibiotic) for seven different aminoglycosides: amikacin (AK 30 μg), gentamicin (CN 30 μg), kanamycin (K 30 μg), neomycin (N 30 μg), netilmicin (NET 10 μg), spectinomycin (SH 25 μg), and tobramycin (TOB 30 μg). Individual data points are presented in Supplementary Fig. 6. Significantly increased growth ($p$-value $< 0.01$, one-sided two-sample t-test) is denoted by an asterisk above the bar, with red asterisks indicating a resistance level beyond the clinical breakpoint (amikacin, gentamicin, tobramycin) or ECOFF (neomycin). Standard deviations are displayed as error bars. Panel **h** shows an overview of the tested antibiotic resistance genes and the aminoglycoside(s) that each gene conferred significantly increased resistance to, with asterisks denoting clinical levels of resistance.

conferred resistance levels above the breakpoints for amikacin (D292, E111, E141, E449, E553, F309), gentamicin (C322, C1264), neomycin (H957), netilmicin (D292, E64, E111, E141, E449, E526, E550, E553, E606, E657, E963, F72, F309, F314) and/or tobramycin (C322, C1264, E449, E553, E963) (Fig. 4, Supplementary Fig. 6).

## Discussion
In this study, we screened ~1 million bacterial genomes and identified genes encoding aminoglycoside-modifying enzymes (AMEs) from 7,613 gene families ($< 70\%$ amino acid identity). This included 50 previously undiscovered AME families co-localizing with genes associated with mobile genetic elements (MGEs) in human pathogens. Genes from 21 of these 50 new pathogen-associated mobile AME families were identified in clinical isolates, demonstrating that they are already circulating among pathogens in the human microbiome. The phenotypes of the predicted AMEs were experimentally confirmed, with 24 of the 28 tested genes (86%) significantly increasing the resistance to at least one aminoglycoside. Our findings extensively expand the number of known clinically relevant AMEs and demonstrate that computational screening can be used to identify potential emerging resistance genes.

Seventeen (61%) of the experimentally validated AMEs conferred resistance above the clinical breakpoints and/or ECOFFs for amikacin, gentamicin, neomycin, netilmicin, and/or tobramycin as defined by EUCAST (Fig. 4, Supplementary Fig. 6). However, it is important to note that extrapolation of exact resistance levels generated in heterologous expression systems should always be done with caution. Also, our assay did not conform exactly to EUCAST standards since the validations used disks containing 30 μg antibiotics (except for netilmicin and spectinomycin, which used disks containing 10 μg and 25 μg respectively), while the EUCAST cut-offs for gentamicin, neomycin, and tobramycin are based on disks with 10 μg loadings[41], and in this sense, the estimates of resistance we present are likely conservative. On the other hand, we used a high expression vector, which likely resulted in higher levels of resistance than what would be observed naturally. Nevertheless, our results show that several of the identified AMEs constitute potential contributors to the declining efficacy of aminoglycosides and could, when present in pathogens and expressed at sufficient levels, prevent successful antibiotic treatment. It should be noted that four of the tested new AMEs did not provide aminoglycoside resistance under our experimental conditions (C24, C118, C715, E18), even after the genes were codon optimized for *E. coli*. It is, however, plausible that these genes are still functional in other hosts or genetic contexts. Interestingly, all four of these AMEs were encoded by genes originally identified in *Streptococcus* sp., indicating that they may be genetically incompatible with *E. coli*. However, two of the other tested AMEs that were associated with *Streptococcus* (E449, E526) did induce a resistance phenotype, demonstrating that at least some AMEs can function in evolutionarily distant hosts.

Our results suggest that the new pathogen-associated AMEs are primarily transmitted via conjugation. Based on strict criteria

(see Methods), as many as 68% of the 50 new pathogen-associated mobile AMEs were co-localized with genes involved in conjugation. This was close to what was observed for the known AMEs, of which 78% were found on conjugative plasmids. We noted that several of the new AMEs were associated with broad-host plasmids, including plasmids carrying type-IV secretion systems of class F and T, which suggests a potential to spread over large evolutionary distances[42]. There were also differences in the genetic context between new and known AMEs. Most striking was the association between AMEs and integrons, where only 6%, compared to 48% of new and known pathogen-associated AMEs, respectively, were located close to class 1 integrases. The new AMEs were also to a lower extent co-localized with known mobile ARGs. Indeed, 91% of the known pathogen-associated AMEs were co-localized with known genes conferring resistance to other antibiotics, including beta-lactams, tetracyclines, and macrolides, which was only true for 54% of the new AMEs. This could indicate that several of the new AMEs found in pathogens represent emerging ARGs that have not yet become associated with the most efficient MGEs and/or co-localized with other more established and well-spread ARGs. Multidrug resistance plasmids carrying large gene arrays are formed by a stepwise process that includes several consecutive evolutionary events distributed over time[43]. A lower tendency for co-localization with other ARGs has also been seen for carbapenemases, of which many are believed to have been transferred into pathogens after the relatively recent introduction of carbapenems for treating bacterial infections[44]. It is thus possible that several of the AMEs identified in this study may become more prevalent over time, and once present in contexts that facilitate transfer and co-selection it cannot be excluded that they may complement, or even replace, some of the well-established aminoglycoside resistance genes.

Several of the new mobile pathogen-associated AMEs identified in this study were present in old samples, with genes from seven families (C1, C4, E18, G5, H100, I80, I269) being carried by bacteria isolated before 1980. These genes have thus been present in pathogens such as *Bacillus anthracis*, *Streptococcus pneumoniae*, and *Salmonella enterica* for at least four decades without becoming sufficiently widespread to be identified as ARGs. Further investigation into these families revealed that each showed a strong preference for a narrow taxonomic host range (*Bacillus spp.*, *Listeria spp.*, *Streptococcus spp.*, *Bacillus spp.*, *Pseudomonas spp.*, *S. enterica*, and *Legionella spp.*, respectively), which suggested that they lack the means to spread efficiently. However, for four of the seven AME families identified in old isolates (C1, E18, G5, I80), a small proportion of genes were present in evolutionarily distant pathogens, indicating horizontal gene transfer (Supplementary Data 3). Among these evolutionarily distant hosts, the earliest confirmed collection date was 2006, and we, therefore, speculate that these genes may have more recently become associated with MGEs that improved their ability to transfer more broadly. It is also possible that these genes, and other AMEs found in this study, are facing strong barriers that hamper their dissemination[45]. Oftentimes, ARGs are associated with an increased fitness cost. This, to a large extent, depends on the gene nucleotide sequence, where the presence of rare codons

in the new host can drastically reduce the translation rate and thereby prevent a sufficiently high expression[46]. It is therefore possible that some of the AMEs identified in this study are not sufficiently beneficial and that most bacteria favour more cost-efficient aminoglycoside resistance determinants[47].

The over 1 million predicted AMEs clustered into almost 8,000 AME families – of which only a very small proportion (1.2%) have been previously characterized. This demonstrates the vast diversity of AMEs, of which many are latently present in many bacterial communities[48]. Indeed, AMEs were ubiquitously present in both environmental and commensal bacteria, including Actinobacteria, Firmicutes, Proteobacteria, Bacteroidetes, Chloroflexi, and Cyanobacteria (Supplementary Fig. 5). Moreover, the genes predicted by the different models showed clear taxonomic preferences, suggesting that they originate from different bacterial phyla (Supplementary Figs. 5, 7)[25]. While most ARGs in non-pathogenic species may not present an immediate threat, it is not unreasonable to assume that some of them could be mobilized and transferred into pathogens in the future. Notably, aside from the 50 new mobile pathogen-associated AME, we identified 433 additional new AME families associated with MGEs, suggesting that the mobilization of several genes may have already occurred (Fig. 1d). This was supported by the phylogenetic analysis which showed clear indications of horizontal transfer events in all trees, suggesting that not only the clinically relevant AMEs are mobile (Supplementary Fig. 5). A highly diverse and partially mobile resistome has recently been described for other ARGs, including beta-lactam, macrolide, tetracycline, and quinoline resistance genes[49–52]. This further emphasizes that the full resistome – including the many ARGs that have not yet been found in pathogens – needs to be considered to fully assess the future risks to human health associated with the promotion of antibiotic resistance.

The identification of new AMEs described in this study was based on computational predictions. These genes should be treated as putative ARGs until they have been experimentally validated. Even if most of the predicted genes were present in monophyletic clades together with known resistance genes (Supplementary Fig. 5), some of them might have recently lost their original functionality. However, the experimental validation showed that 86% of the tested genes provided increased resistance in *E. coli* to at least one aminoglycoside. This is in line with previous studies using the same methodology, for which 60-90% of tested genes produced the expected phenotype[49–52]. This study thus provides further evidence that computational prediction can be used to expand the resistome beyond the limited number of gene sequences that are currently present in the ARG databases. We also argue, based on our results, that computational screening of bacterial isolates constitutes the most efficient large-scale approach to identifying new, potentially emerging, resistance determinants.

In this study, we identified an abundance of new mobile AMEs carried by pathogenic species, showing that computational methods enable the identification of potentially emerging ARGs before they become widespread. Experimental validation showed high accuracy for predicting functional new resistance determinants, and several of the tested new AMEs provided resistance well beyond clinical breakpoints. The introduction of new and potent ARGs constitutes a serious threat to the efficacy of existing and future antibiotics[1–3]. Knowledge about which ARGs may become a clinical problem in the future is crucial for the implementation of suitable preventive measures to limit dissemination. It is also vital for sequencing-based methodologies that today are routinely used in diagnostics[53], infection control[54], and surveillance[55]. Indeed, the results of this study show that existing ARG databases are clearly limited and will only provide information about a small part of the clinically relevant resistome. In fact, this single study almost doubles the number of known *aac* and *aph* gene variants circulating in pathogens. It is thus imperative that the reference databases of ARGs are expanded and here computational prediction offers an accurate and scalable method that can complement the more traditional approaches.

## Methods

**Model creation and optimization**. Nine profile hidden Markov models (HMMs) were created and optimized using fARGene v0.1[35] as follows. Nucleotide sequences representing *aac* and *aph* genes were downloaded from ResFinder v4.0[20] and translated using EMBOSS Transeq v6.5.7.0[56]. Since minimal differences are required for some aminoglycoside resistance genes to be considered different variants[57], the protein sequences were clustered at 90% amino acid identity using USEARCH v8.01445 with parameters '-cluster_fast -id 0.9'[58] to reduce redundancy.

The centroid sequences representing GNAT AACs, non-GNAT-like AACs, and APHs were aligned separately using mafft v7.23[59], with default parameters. Phylogenetic trees were created from the alignments using FastTree v2.1.10[60], with default parameters. From the resulting trees, nine subsets of sequences that clustered together were identified (Supplementary Figs. 1, 2, 3). These subsets were used to create nine HMMs using 'fargene_model_creation' from fARGene v0.1[35]. Of these models, six (denoted A-F) represented AACs (with model C representing non-GNAT-like AAC(3)s), and three (denoted G-I) represented APHs (Supplementary Data 1). Sequences that deviated from the nine subsets could not be included in any model without severely reducing its performance and were therefore excluded.

For each model, the sensitivity was estimated using leave-one-out cross-validation. The specificity was estimated using a negative set of protein sequences that are evolutionarily close to the AME but have not been associated with aminoglycoside resistance. For the six AAC models, the specificity was estimated from a set of 374 sequences representing evolutionarily related acetyltransferases, including members of the GNAT family of N-acetyltransferases to which most AAC enzymes also belong. For the three APH models, 60 sequences representing homologs of homoserine kinase II and macrolide phosphotransferases were used to estimate the specificity (Supplementary Data 1). For all models, domain score thresholds were assigned with the criteria that both sensitivity and specificity should be as high as possible. However, to ensure a low false positive rate, high specificity took priority over high sensitivity. Additionally, throughout the model creation process, we paid attention to any overlap between the sequences predicted by different models, and we found no such overlap for the final models.

**Resistance gene prediction and phylogenetic analysis**. All bacterial genomes from NCBI Assembly (downloaded August 2021) were searched for genes encoding AMEs with fARGene v0.1[35] using the new HMMs. For each model, AME families were created by clustering the protein sequences, together with their corresponding reference sequences from ResFinder, at 70% amino acid identity using USEARCH v8.0.1445[58] with parameters '-cluster_fast -id 0.7'. Nine phylogenetic trees were created from the representative centroid sequences using the methodology described in the previous section. For each centroid, the closest known homolog was identified by applying BLASTx v2.10.1[61], using a custom database primarily based on ResFinder v4.0[20] but augmented with sequences from CARD[62]. The obtained identity score was visualized together with the trees using ggtree v3.0.4[63].

**Genetic context analysis**. For each of the 7612 predicted AME families, genetic regions of up to 10 kb up- and downstream of the genes in the cluster were retrieved using GEnView v0.1[64]. These genetic contexts were screened for MGEs, including conjugative elements, integrons and insertion sequences (ISs), and known ARGs. Conjugative elements were identified by translating the genetic regions in all six reading frames using EMBOSS Transeq v6.5.7.0[56] and screening the translated contexts with 124 HMMs from MacSyfinder CONJScan v2.0[65], using HMMER v3.1b2[66]. Integron Finder v1.5.1[67] was applied to the genetic contexts to identify integrons. insertion sequences and co-localized mobile ARGs were identified by applying BLASTx v2.10.1[61] to the genetic contexts. For insertion sequences, a reference database based on ISFinder[68,69] was used to find the best among overlapping hits located within 1 kb of the predicted AME, with the criteria of hits showing >50% coverage and >90% nucleotide identity. For co-localized ARGs, ResFinder v4.0 was used as a reference database[20], with the criterion of hits showing >90% nucleotide identity.

The similarity between the proteins encompassing each family and known AMEs was evaluated using BLASTx v2.10.1[61] and the same database as during the phylogenetic analysis. Each AME family was classified as either known or new based on the highest obtained identity score, where >70% amino acid identity to any ResFinder/CARD AME was considered known. Additionally, pathogenic host species within each cluster were identified by cross-referencing the represented species with a reference list based on the PATRIC database[70]. Finally, for each AME family, metadata about the host genome isolates was collected from NCBI using Entrez Direct v13.9[71,72].

**Experimental validation**. Representative genes from 34 new AME families were selected for functional validation. Together with four positive (*aph(9)-Ia* [U94857], *aac(3)-IIb* [M97172], *aph(3')-III* [M26832], *aph(3″)-Ib* [M28829]) and two negative (*tet(A)* [AF534183], *dfrA1* [FJ591049]) control gene sequences obtained from CARD[62], the genes were codon optimized for *E. coli* using the GenScript online codon optimization tool[73] and synthesized by Twist Bioscience (USA). These genes were shipped pre-inserted within the pBAD multiple cloning site region of the pBAD/myc-His B plasmid vector (Invitrogen, USA). Upon arrival, the lyophilized DNA was resuspended in nuclease-free water to a final concentration range of 10–20 ng/µl (validated using a Qubit 2.0 Fluorometer [Invitrogen, USA]) and stored at −20 °C.

Approximately 15 ng of each DNA sample was mixed with 50 µl of electrocompetent *E. coli* TOP10 cells and transferred to chilled 1 mm Electroporation Cuvettes (Bio-Rad, USA). Electroporation was performed at 1.7 kV for 5.3 ms using the Eppendorf™ Eporator™ (Eppendorf, Germany). The transformed cells were immediately resuspended in 1 ml preheated SOC medium and incubated for 1 hour at 37 °C while shaking at 300 rpm. The transformants were plated on LB agar with and without supplemented 100 µg/ml ampicillin and incubated overnight at 37 °C. Successful transformants were identified as single colonies on plates supplemented with ampicillin. The no-plasmid control indicated no growth on these plates, confirming the presence of plasmids in the cloned samples. A single colony from each transformed sample was streaked on LB agar plates supplemented with 100 µg/ml ampicillin, incubated overnight at 37 °C, and stored at 4 °C the following day.

Validation of the predicted AMEs was performed using the disk diffusion test[74]. Eight aminoglycosides were selected (kanamycin [K 30 µg], gentamicin [CN 30 µg], neomycin [N 30 µg], spectinomycin [SH 25 µg], netilmicin [NET 10 µg], amikacin [AK 30 µg], tobramycin [TOB 30 µg], streptomycin [S 10 µg], together with two negative controls (tetracycline [TE 30 µg] and trimethoprim [W 5 µg] [Oxoid, United Kingdom]). A single colony from each of the transformed clones was inoculated in LB broth supplemented with 100 µg/ml ampicillin and incubated overnight at 37 °C while shaking at 150 rpm. The following day 100 µl of the overnight culture was inoculated in fresh LB broth supplemented with 100 µg/ml ampicillin and incubated at 37 °C for ~6 h. L-(+)-arabinose (Sigma, USA) was added to each broth (0.1%) and incubation continued until an $OD_{600}$ measurement of 0.7 was reached. Each of the samples was then spread on Mueller-Hinton agar plates supplemented with 0.1 % L-(+)-arabinose. One of each antibiotic disk was added to the plates using an antibiotic disk dispenser (Oxoid, United Kingdom). Plates were incubated at 30 °C, thereby avoiding the formation of inclusion bodies[75], for ~18 h. Results were taken by measuring the inhibition zones of each antibiotic on the various samples. The disk diffusion screening was repeated for a total of three replicates. Confirmation of the cloned inserts used in the expression studies was validated using Sanger Sequencing.

Since the *E. coli* TOP10 strain is resistant to streptomycin, we excluded six tested AMEs that were only expected to confer resistance to this aminoglycoside (I26, I64, I81, I91, I270, I642), based on the resistance profile associated with their respective closest known homolog. The inhibition zone diameters produced by these AMEs for the other tested aminoglycosides are reported in the supplementary material (Supplementary Fig. 6, Supplementary Data 5).

**Statistics and reproducibility**. Phylum enrichment analysis was performed to test whether the AMEs predicted by different models had specific taxonomic affiliations. For each model, the number of unique host species from each of the main four bacterial phyla (Actinobacteria, Bacteroidetes, Firmicutes, and Proteobacteria) was counted and compared to the number of species from the same phylum present in the database using Fisher's exact test. A $p$-value < 0.01 was considered significant.

To determine if the experimentally tested AMEs conferred significantly increased resistance in *E. coli*, $p$-values were calculated using a one-sided two-sample t-test. For each gene, the inhibition zone diameters obtained from the three replicates were tested against the diameters from the corresponding negative controls (both empty plasmid and no-plasmid controls) for each tested aminoglycoside. For significant tests ($p$-value < 0.01), the average inhibition zone diameter of the replicates was compared to EUCAST's clinical breakpoints (amikacin, gentamicin, tobramycin) or epidemiological cut-off values (ECOFFs; neomycin)[41] when available. Genes were considered to give clinical resistance when the average zone diameter plus one standard deviation was below the corresponding clinical breakpoint/ECOFF.

**Reporting summary**. Further information on research design is available in the Nature Portfolio Reporting Summary linked to this article.

## Data availability

All genomes used for this study are publicly available at https://www.ncbi.nlm.nih.gov/assembly. The models created and used in this study are available from https://github.com/fannyhb/fargene. Protein sequences, nucleotide sequences, and metadata corresponding to the new genes predicted in this study are listed in Supplementary

Data 2. Raw data from the disk diffusion tests is presented in Supplementary Data 5. Source data used to generate the figures can be found in Supplementary Data 6.

## Code availability

The code used for the genetic context analysis is available at https://github.com/davidgllund/ARG_context_analysis_pipeline[76].

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

## Acknowledgements

This research was supported by the Swedish Research Council (VR) (2018–02835, 2018-05771 and 2019–03482). Funding sources took no part in the design, analysis, or interpretation of the results.

## Author contributions

D.L., F.B., A.J., D.G.J.L., and E.K. designed the study and developed the approach. D.L. created and optimized the probabilistic models. M.P.-M. collected the data and ran the fARGene analysis. D.L. implemented the computational analysis pipeline, including phylogenetic analysis and genetic context analysis. R.D.C. and C.-F.F. performed the experimental validation. All authors discussed the results and their implications. D.L. and E.K. drafted the manuscript. All authors edited and approved the final manuscript.

## Funding

## Competing interests

The authors declare no competing interests.
