## [Peer Review File · Communications Biology]

Reviewers' comments:

Reviewer #1 (Remarks to the Author):

Summary: Kristiansson et al. present a computational screen of bacterial genomes to identify novel AMEs. The significance of this work is described perfectly, and the authors clearly address the need for tools to discover ARGs prior to clinical prevalence. Utilizing a tool previously developed by this research group, the analysis is presented as broad strokes and lacks clear examples of novel AMEs found and how they compare to current clinical threats. The authors put an incredible amount of effort into the analysis of their dataset, breaking down the over 1 million predicted genes into many different categories for further analysis. i.e. species of origin if known, pathogenic vs environmental. However, quantities for different groupings are reported with a lack of detail provided for key findings within the dataset. This leaves the reader to sort through the extensive supplementary data. Below are comments to aid in producing a clearer narrative for a second round of review.

Main concerns:

- The class of aminoglycosides are a critically important class of antibiotics with next-generation scaffolds currently in development to evade resistance. It is unfortunate that there was no inclusion of the parent compounds for functional assays – apramycin and paromomycin – to identify emerging clinical threats.
- The strength in reviving this class of antibiotics to combat AMR in part lies in the structural diversity and opportunities for semi-synthesis. The authors didn't provide any introductory figure presenting the aminoglycoside structures or means of inactivation. This limits the accessibility/understanding of manuscript content and appreciation for the choice of drug class the authors investigate.
- The authors incorrectly imply that all AACs belong to the GNAT superfamily of proteins. In addition, there is no sequence published for AAC(1) and thus it's protein familial status cannot be known. The accurate family assignment and terminology for AMEs – and other resistance mechanisms – is critical for the successful curation or development of identification/prediction computational models for emerging ARGs. Phylogeny presented for the AACs wrongly includes AAC(3)s that are distinct from AAC(3)-I GNATs at a sequence and structural level. AAC(3)-XI is the only other GNAT protein of the AAC(3)s based on sequence. The authors highlight the need for knowledge about evolutionary processes but miss the mark on the background information of AMEs for their data to be utilized appropriately and accurate model construction. On a related note – can the authors comment on the lack of isoform designation for AAC(3)-I in the clade identified as model A in Figure S1?
- The resistance phenotypes to differentiate AMEs are represented by a roman numeral. The authors presented susceptibility results in Figures 4 and S5 referencing the model grouping but no clear discussion of a unique substrate specificity profile. Are these proteins within a group just all variants capable of the same resistance phenotype?

Formatting comments and suggestions

The authors need to reference Supplementary Figures 1 and 2 in the main manuscript, it is unclear until the methods section as to how the models were generated if not familiar with bioinformatics.

Line 118 – ACCs should read AACs.

Line 130 – 7,613 families? Groups or variants may be a better phrasing to avoid confusion? This applies to all instances throughout manuscript, clarity from the authors is needed.

Line 138 – AME class E [AAC(6')] versus Model E [(AAC(6'))] in line 121, selection of a single format is needed.

This will need to be reformatted to improve flow of text and prevent confusion. The authors' use of "class" and "family" throughout the text also creates confusion due to the clash of terminology meaning in other areas of biochemistry.

Reviewer #2 (Remarks to the Author):

In their manuscript "Extensive screening reveals previously undiscovered aminoglycoside resistance genes in human pathogens" Lund et al. use extensive bioinformatic screening to predict unprecedentedly high numbers of previously undiscovered aminoglycoside resistance genes from existing whole genome sequences. They thereafter aim at verifying that the predicted genes are indeed conferring resistance by cloning a small subset of them into an expression vector of *E. coli*.

I believe that the research topic of finding novel resistance genes merits investigation and that more inclusive databases of potential resistance genes are of value for future identification of resistance determinants in bacteria. The manuscript as such is well written and fairly easy to understand.

With regards to the used methodology, the bioinformatics pipeline used and the analysis carried out is overall convincing and provides promising results, to which I have only minor additional questions. Furthermore, the number of bacterial genomes screened is impressive. However, the experimental approach for verification has several flaws that need to be addressed prior to a potential publication.

Please find below my major and minor comments for improving the manuscript:

1) I am slightly surprised by the high claimed number of novel aminoglycoside resistance genes (>1,000,000) detected in this study. The authors cluster them by protein sequences to obtain ~30k unique protein sequences. Consequently, the statement should be amended to say that >1,000,000 hits of potential resistance genes were generated and that of these a high number are due to hits for the identical gene in different whole genomes of different bacteria. Please clarify this throughout the manuscript to avoid potential overstatements of the actually detected genes.

2) The predicted resistance genes originate from 9 individual fARGene runs based on models generated with different training sets. Is it possible that the same ARG within a bacterium is detected by several of these 9 fARGene runs and that this leads to an artificial inflation of hits? This would explain the extremely high number of hits/bacterium of >1. Furthermore, are those genes that constitute the training set for one model correctly identified by using one or more of the other 8 models?

3) I agree that geography, collection date and isolation source are relevant when assessing the spread of the putative genes. For individual genes observed this is highly valuable. However, I wonder if in a significantly uneven dataset like this one any interpretation of the overall data (non-normalized numbers) regarding observed ame families (e.g. figure 3) provides any reliable information. I would hence suggest to omit these parts.

4) Regarding the experimental validation a number of methodological and interpretative flaws need to be addressed. First, the authors claim resistance if significantly smaller inhibition zones are detected. However, for all antibiotics without defined clinical breakpoints, and all those denominated with black stars in Figure 4, exclusively decreased susceptibility was detected. The claim of resistance is here a clear overstatement. Consequently, resistance was exclusively detected for 10 of the tested 28 genes rather than the far more impressive sounding 24. While the other genes seem to have certain effects, the claimed resistance was not demonstrated and seems to be assigned arbitrarily.

5) The claimed resistance for those genes that confer effects above the clinical breakpoint seems especially concerning as the authors are using a high expression vector to express the genes in *E. coli*. If even under such a strong promoter no clinically relevant effect can be observed the ability to confer resistance under “more natural” conditions with likely a weaker promoter should be questioned.

6) I am wondering why disks with significantly higher antibiotic loads compared to those standardized for the EUCAST assay were used. I agree that anything below clinical breakpoint for the lower concentrations can be considered resistant, however, anything above clinical breakpoint could equally be resistant according to the standardized protocol, but simply not detected as resistant due to the higher antibiotic load. This significantly complicates the discussion of these results and needs to be at least addressed.

7) For Amikacin no positive control (e.g. AAC(6')-Ib) was carried out. Instead APH(3')-Ib was

chosen as a positive control gene, but it doesn't confer resistance to any of the chosen antibiotics.

8) The discussion of origin of the putative resistance genes (with regard to training sets) confers resistance to which of the tested antibiotics is missing crucial points: For example, why are the majority of genes displaying a significant effect for Kanamycin susceptibility originating from the AAC training sets, while none are found for one of the APH3 training sets, despite aph3 conferring the highest level of effect among the positive controls?

Response to review comments.

Reviewer 1

Summary: Kristiansson et al. present a computational screen of bacterial genomes to identify novel AMEs. The significance of this work is described perfectly, and the authors clearly address the need for tools to discover ARGs prior to clinical prevalence. Utilizing a tool previously developed by this research group, the analysis is presented as broad strokes and lacks clear examples of novel AMEs found and how they compare to current clinical threats. The authors put an incredible amount of effort into the analysis of their dataset, breaking down the over 1 million predicted genes into many different categories for further analysis. i.e. species of origin if known, pathogenic vs environmental. However, quantities for different groupings are reported with a lack of detail provided for key findings within the dataset. This leaves the reader to sort through the extensive supplementary data. Below are comments to aid in producing a clearer narrative for a second round of review.

Reply: We thank the reviewer for the constructive comments. This study aims to systematically screen large datasets for new resistance genes that may constitute potential threats to human health. Since this is a significant undertaking and, due to the large number of genes that we were able to identify, describing them all in detail is simply not possible in a single study. We, thus, see a more detailed characterization of each of the predicted genes as an important topic for future research. As the reviewer correctly points out, we have provided detailed information in the supplementary material to facilitate future studies.

In addition to implementing the results according to the reviewer's suggestions, we have also updated the results with newly released EUCAST resistance cut-offs for netilmicin and neomycin. This resulted in an additional seven genes producing resistance exceeding the breakpoints.

Main concerns:

1. *Comment:* The class of aminoglycosides are a critically important class of antibiotics with next-generation scaffolds currently in development to evade resistance. It is unfortunate that there was no inclusion of the parent compounds for functional assays – apramycin and paromomycin – to identify emerging clinical threats.

Reply: When testing the functionality of the new AMEs we screened for resistance against 8 different aminoglycosides primarily selected based on clinical relevance. We agree with the reviewer that it would be of interest to examine the complete resistance profile of the newly discovered AMEs, however, due to the large number of genes, complete characterization is not feasible to do within a single study. In addition to testing the susceptibility against more compounds, we also argue that the genes need to be assessed in more hosts than *E. coli* to properly assess their clinical impact. Since both these aspects require significant undertakings, we see them as beyond the scope

of this study and, thus, future research topics.

2. *Comment:* The strength in reviving this class of antibiotics to combat AMR in part lies in the structural diversity and opportunities for semi-synthesis. The authors didn't provide any introductory figure presenting the aminoglycoside structures or means of inactivation. This limits the accessibility/understanding of manuscript content and appreciation for the choice of drug class the authors investigate.

Reply: The structure of aminoglycosides and mechanisms of inactivation associated with different AMEs have been extensively documented in the literature. To avoid repeating already published results, we decided to not include any introductory figure but instead cite several relevant sources in the Introduction.

3. *Comment:* The authors incorrectly imply that all AACs belong to the GNAT superfamily of proteins. In addition, there is no sequence published for AAC(1) and thus its protein familial status cannot be known. The accurate family assignment and terminology for AMEs – and other resistance mechanisms – is critical for the successful curation or development of identification/prediction computational models for emerging ARGs. Phylogeny presented for the AACs wrongly includes AAC(3)s that are distinct from AAC(3)-I GNATs at a sequence and structural level. AAC(3)-XI is the only other GNAT protein of the AAC(3)s based on sequence. The authors highlight the need for knowledge about evolutionary processes but miss the mark on the background information of AMEs for their data to be utilized appropriately and accurate model construction.

On a related note – can the authors comment on the lack of isoform designation for AAC(3)-I in the clade identified as model A in Figure S1?

Reply: The reviewer is correct in that the AAC(3) proteins used to construct model C do not belong to the GNAT family. This has been corrected in the manuscript (Lines 53-55), and the phylogenies presented in the supplementary material have been revised. Specifically, the non-GNAT AAC(3) sequences have been removed from Supplementary Figure 1, and a new supplementary tree has been added consisting only of these sequences, using yokD, an AAC(3) homolog from phages with questionable functionality (<https://www.nature.com/articles/ismej201690>), as outgroup. The new Supplementary Figures 1 and 2 are presented below.

Supplementary Figure 1. Phylogenetic tree displaying known AAC sequences from the GNAT-family of N-acetyltransferases. The tree was built from centroid sequences after clustering at 90% amino acid identity. The five groups of sequences that were used to create separate models A–B, D–F are marked in the tree.

Supplementary Figure 2. Phylogenetic tree displaying known, non-GNAT like AAC(3) sequences. The tree was built from centroid sequences after clustering at 90% amino acid identity. The sequences that were used to create model C are marked in the tree.

Due to the lack of homologs to AAC(3) in sequence repositories, we could not compile a suitable negative dataset and, therefore, used a significance threshold score that was similar to the other models.

Relating to the last point, the sequences and nomenclature were taken verbatim as they appear in the ResFinder database. According to the accession number associated with the gene (AJ877225), it is a gentamicin resistance gene identified in *Pseudomonas aeruginosa*. However, since the sequence does not adhere to standard nomenclature, nor was it included in the creation of any model, we have removed it from the revised Supplementary Figure 1.

4. *Comment:* The resistance phenotypes to differentiate AMEs are represented by a roman numeral. The authors presented susceptibility results in Figures 4 and S5 referencing the model grouping but no clear discussion of a unique substrate specificity profile. Are these proteins within a group just all variants capable of the same resistance phenotype?

Reply: To assign a unique resistance profile – i.e., a Roman numeral – to the tested new genes requires a more extensive screening using more aminoglycosides. While the resulting phenotype of a new resistance gene can be somewhat anticipated based on its closest known homolog, it is not possible to be certain without experimental validation. Similarly, we would expect AMEs from the same new gene family to produce the same phenotype, but we cannot know without further testing. In fact, even with extended susceptibility testing, we would not be able to conclude whether the new AMEs are acting at the same site as their closest known homolog. However, to clarify the anticipated resistance profiles of the genes encompassing each AME class, we have included more detailed information about the included reference phenotypes in Table 1 and Supplementary Tables 1 and 2.

1. *Comment:* The authors need to reference Supplementary Figures 1 and 2 in the main manuscript, it is unclear until the methods section as to how the models were generated if not familiar with bioinformatics.

Reply: Changes have been made according to the reviewer's suggestion.

2. *Comment:* Line 118 – ACCs should read AACs.

Reply: The typo has been corrected.

3. *Comment:* Line 130 – 7,613 families? Groups or variants may be a better phrasing to avoid confusion? This applies to all instances throughout manuscript, clarity from the authors is needed.

Reply: “Family” is the most common way to refer to a group of related proteins and is used as nomenclature by central sequence repositories such as InterPro and Pfam (now integrated into InterPro). By contrast, “variants” is commonly used to describe smaller genetic variations, and “groups”, in our opinion, is too general a term. We, therefore, argue that keeping the standard notation of “family” for a cluster of proteins is most appropriate.

4. *Comment:* Line 138 – AME class E [AAC(6')] versus Model E [(AAC(6'))] in line 121, selection of a single format is needed.

This will need to be reformatted to improve the flow of text and prevent confusion. The authors' use of “class” and “family” throughout the text also creates confusion due to the clash of terminology meanings in other areas of biochemistry.

Reply: We agree with the reviewer that our nomenclature could cause unnecessary confusion, and we have therefore changed all instances of “AME class X” to “genes predicted by model X” throughout the manuscript.

Reviewer 2

In their manuscript “Extensive screening reveals previously undiscovered aminoglycoside resistance genes in human pathogens” Lund et al. use extensive bioinformatic screening to predict unprecedentedly high numbers of previously undiscovered aminoglycoside resistance genes from existing whole genome sequences. They thereafter aim at verifying that the predicted genes are indeed conferring resistance by cloning a small subset of them into an expression vector of *E. coli*.

I believe that the research topic of finding novel resistance genes merits investigation and that more inclusive databases of potential resistance genes are of value for future identification of resistance determinants in bacteria. The manuscript as such is well written and fairly easy to understand.

With regards to the used methodology, the bioinformatics pipeline used and the analysis carried out is overall convincing and provides promising results, to which I have only minor additional questions. Furthermore, the number of bacterial genomes screened is impressive. However, the experimental approach for verification has several flaws that need to be addressed prior to a potential publication.

Please find below my major and minor comments for improving the manuscript:

1. *Comment:* I am slightly surprised by the high claimed number of novel aminoglycoside resistance genes (>1,000,000) detected in this study. The authors cluster them by protein sequences to obtain ~30k unique protein sequences. Consequently, the statement should be amended to say that >1,000,000 hits of potential resistance genes were generated and that of these a high number are due to hits for the identical gene in different whole genomes of different bacteria. Please clarify this throughout the manuscript to avoid potential overstatements of the actually detected genes.

Reply: While we appreciate the reviewer's comment, we would like to clarify that only ~15% of the ~1,000,000 detected ARGs were novel. Instead, most represented known variants that are common in pathogens, either on plasmids (where it is not uncommon for multiple AMEs to appear on the same plasmid) or as part of the chromosome (for example, *Salmonella enterica*, which made up a substantial portion of the genome database, is known to carry the chromosomal *aac(6')-Iy* gene). This largely explains the large number of predicted genes. We have edited the manuscript to clarify this issue by specifying that the 1,071,815 predicted genes encoded 34,053 unique proteins (Line 24, 101, 130).

2. *Comment:* The predicted resistance genes originate from 9 individual fARGene runs based on models generated with different training sets. Is it possible that the same ARG within a bacterium is detected by several of these 9 fARGene runs and that this leads to an artificial inflation of hits? This would explain the extremely high number of hits/bacterium of >1. Furthermore, are those genes that constitute the training set for one model correctly identified by using one or more of the other 8 models?

Reply: When creating the models, we took great care to ensure that the overlap between them in terms of identified sequences was minimal. In the case of the dataset used for this study, we found no overlap among the sequences predicted by each model. Of course, this also means that no model was able to successfully detect any reference genes from the training set of another model. We now clarify this in the manuscript (Line 357-359)

3. *Comment:* I agree that geography, collection date and isolation source are relevant when assessing the spread of the putative genes. For individual genes observed this is highly valuable. However, I wonder if in a significantly uneven dataset like this one any interpretation of the overall data (non-normalized numbers) regarding observed ame families (e.g. figure 3) provides any reliable information. I would hence suggest to omit these parts.

Reply: The reviewer is right that this data is very biased, which is why we have not performed any statistics on it. We have removed most of the observations made about geography to avoid over-interpretation (Line 175-181, original manuscript), however, we still argue that there is value in the observation that genes have occurred in different environments, locations, and times at least once, even if no definite conclusions can be drawn from it.

4. *Comment:* Regarding the experimental validation a number of methodological and interpretative flaws need to be addressed. First, the authors claim resistance if significantly smaller inhibition zones are detected. However, for all antibiotics without defined clinical breakpoints, and all those denominated with black stars in Figure 4, exclusively decreased susceptibility was detected. The claim of resistance is here a clear overstatement. Consequently, resistance was exclusively detected for 10 of the tested 28 genes rather than the far more impressive sounding 24. While the other genes seem to have certain effects, the claimed resistance was not demonstrated and seems to be assigned arbitrarily.

Reply: The experimental validation aims to investigate if the predicted genes induce a phenotype with reduced susceptibility compared to wild-type bacteria – i.e. using the microbiological definition of resistance – and the primary focus was not on clinically relevant levels of resistance. Indeed, defining resistance as “decreased susceptibility”, is very common in AMR research that deals with questions that reach beyond the clinical domain. Indeed, there is a strong rationale for choosing the microbiological definition over the clinical definition in this type of research. Firstly, extrapolating an exact level of resistance provided by a gene artificially introduced in an expression vector to the real world – where a resistance gene is located in its natural context – is very difficult for numerous reasons. The resistance levels associated with a gene in its natural context could certainly be both higher and lower than in the expression system. Secondly, the microbiological definition is less claiming in that it does not involve a breakpoint, just a clear increase in resistance compared to the control bacteria not expressing the gene. Having said that, we highlight the genes demonstrating a resistance level above the clinical breakpoints in the manuscript since it says something about the potential of a gene.

To make these distinctions clear, we have clarified our definition of resistance (Line 189) and verified that we use the term “clinical resistance” only when the measured phenotypes are compared to clinical breakpoints.

5. *Comment:* The claimed resistance for those genes that confer effects above the clinical breakpoint seems especially concerning as the authors are using a high expression vector to express the genes in *E. coli*. If even under such a strong promoter no clinically relevant effect can be observed the ability to confer resistance under “more natural” conditions with likely a weaker promoter should be questioned.

Reply: The reviewer is correct that this setup is not natural, which we now also clarify in the text (Line 220-221). However, genes can and are referred to as “antibiotic resistance genes” even if they often do not provide resistance above the clinical breakpoint in either their natural context or when overexpressed in a vector (e.g. the quinolone resistance

genes, *qnr*). This nomenclature is also applied within the clinical community. Second, genes do not become clinically irrelevant just because they, by themselves, do not bring the host to a level of resistance above the clinical breakpoint. For example, the clinically problematic carbapenemases of the OXA family cause clinical levels of resistance only in combination with e.g. porin mutations.

Our choice of a high-expression vector relates to our choice to apply the microbiological definition of resistance. The key aim is to investigate if the predicted genes can increase resistance above that of wild-type bacteria, not to define the exact level of resistance provided. By showing that a large proportion of the experimentally tested genes provide a functional mechanism that can contribute to the protection against antibiotics, we show that the models can do accurate predictions. Note also that most of the tested genes are not adapted for *E. coli* (although we tried to make them more compatible by codon optimization). Indeed, some of the genes that did not induce a resistant phenotype *in E. coli* could very well be more problematic in other species.

6. *Comment:* I am wondering why disks with significantly higher antibiotic loads compared to those standardized for the EUCAST assay were used. I agree that anything below clinical breakpoint for the lower concentrations can be considered resistant, however, anything above clinical breakpoint could equally be resistant according to the standardized protocol, but simply not detected as resistant due to the higher antibiotic load. This significantly complicates the discussion of these results and needs to be at least addressed.

Reply: Note that our primary aim is to investigate increases in resistance rather than defining absolute levels of resistance, in which case the absolute content of the discs is less important. Still, we agree with the reviewer that our results are likely conservative given that a higher antibiotic load was used compared to the EUCAST standards. Furthermore, we chose to use a single host for the experimental validation, so whether the resistance phenotypes are weaker or, indeed, stronger in other pathogens is unknown. This was briefly mentioned before in the Discussion, but we have now expanded this part by elaborating on the limitations in extrapolating resistance levels from heterologous expression systems to resistance levels in the genes' natural contexts (Line 213-221).

Please note that we have also updated the results with newly released EUCAST resistance cut-offs for netilmicin and neomycin. This resulted in an additional seven genes producing resistance exceeding the breakpoints. The updated Figure 4 and Supplementary Figure 6 are presented below.

Fig. 4. Results from disk diffusion tests using *E. coli* and 28 selected new AMEs. Panels a-g show the mean inhibition zone diameter difference [mm] between clones carrying new AMEs (three replicates for each gene and antibiotic) and susceptible controls for seven different aminoglycosides: amikacin (AK 30 μ g), gentamicin (CN 30 μ g), kanamycin (K 30 μ g), neomycin (N 30 μ g), netilmicin (NET 10 μ g), spectinomycin (SH 25 μ g), and tobramycin (TOB 30 μ g). Significantly increased growth (p -value<0.01, one-sided two-sample t-test) is denoted by an asterisk above the bar, with red asterisks indicating a resistance level beyond the clinical breakpoint (amikacin, gentamicin, tobramycin) or ECOFF (neomycin). Standard deviations are displayed as error bars. Panel h shows an overview of the tested antibiotic resistance genes and the aminoglycoside(s) that each gene conferred significantly increased resistance to, with asterisks denoting clinical levels of resistance.

Supplementary Figure 6. Mean zone diameters [mm] observed during disk diffusion tests of predicted new aminoglycoside resistance genes in *E. coli*. The mean diameters were calculated from three replicates per tested gene and antibiotic. Standard deviations are displayed as error bars. Clinical breakpoints and/or epidemiological cut-off values (ECOFFs) taken from the EUCAST database (when available) are represented as dashed black lines. **a** Amikacin [30 µg]. **b** Gentamicin [30 µg]. **c** Kanamycin [30 µg]. **d** Neomycin [30 µg]. **e** Netilmicin [10 µg]. **f** Spectinomycin [25 µg]. **g** Tobramycin [30 µg].

7. *Comment:* For Amikacin no positive control (e.g. AAC(6')-Ib) was carried out. Instead APH(3')-Ib was chosen as a positive control gene, but it doesn't confer resistance to any of the chosen antibiotics.

Reply: Among the selected positive control genes, APH(3')-III is reported to confer resistance to Amikacin according to several sources including ResFinder (https://bitbucket.org/genomicepidemiology/resfinder_db/src/master/phenotypes.txt), CARD (<https://card.mcmaster.ca/ontology/39047>), and Shaw et al. 1993 (<https://journals.asm.org/doi/abs/10.1128/mr.57.1.138-163.1993>), and therefore this gene was the intended positive control. However, while the gene successfully conferred resistance to Neomycin and Kanamycin it unexpectedly was not able to confer measurable resistance to Amikacin under our experimental setup. APH(3'')-Ib was included as a positive control against Streptomycin, however, for reasons explained in the paper those experiments were unsuccessful. Since all other positive controls behaved as expected, we see no indication that our experimental assay should be unreliable. It should also be emphasized that only the negative controls were used for calculating whether a tested gene conferred significantly increased resistance to any given aminoglycoside, so the lack of a positive control for amikacin does not affect the presented results.

8. *Comment:* The discussion of origin of the putative resistance genes (with regard to training sets) confers resistance to which of the tested antibiotics is missing crucial points: For example, why are the majority of genes displaying a significant effect for Kanamycin susceptibility originating from the AAC training sets, while none are found for one of the APH3 training sets, despite aph3 conferring the highest level of effect among the positive controls?

Reply: The reviewer is correct in that none of the tested genes predicted by Model I (which includes APH(3')-IIc in its training data) produced a Kanamycin resistant phenotype. However, this model is primarily trained to predict APH(6)-I homologs, which are not known to confer resistance to Kanamycin. Although some gene variants in three of the AME families selected for testing (I63, I80, I641), most closely resembled APH(3')-IIc based on sequence identity, the observed similarities were not very high. Indeed, these AME families also included genes that were more similar to APH(6)-I based on sequence identity. Since all the gene variants selected for the validation experiments belong to the latter category, we did not expect them to produce an APH(3')-II phenotype.

Furthermore, we argue that it is not relevant to look at the absolute number of tested AAC genes (i.e. predicted using Model A–F) associated with significantly increased Kanamycin resistance since the number of tested AAC genes was larger than the number of tested APH genes (23 vs 12). If we instead compare the proportions of tested genes that conferred significantly increased to Kanamycin, we can see that 100% of the tested genes predicted by model H (all of which were indicated to be APH(3') homologs based on sequence identity) conferred significantly increased resistance to Kanamycin. Indeed, 3 of the 4 tested genes predicted by model H resulted in Kanamycin resistance levels comparable to the positive control.

REVIEWERS' COMMENTS:

Reviewer #1 (Remarks to the Author):

Summary: In their revised manuscript, the authors made significant improvements to the text which presents a much clearer narrative. Following the suggestions from both reviewers, the authors clearly define terms and provide additional details for their analyses to prevent over-interpretation of the data. The authors end their manuscript with a thought-provoking discussion and provide directions for future studies.

Minor notes

(1) Appropriate corrections have been made to the phylogenies and nomenclature adjustments for model references prevent confusion. These changes add to the manuscript's potential impact, improving surveillance for a class of modifying enzymes often faced with annotation inconsistencies in public databases. Great improvements were made on the presentation of phylogenies in supplementary data figures, easier to read.

- Minor suggestion would be to find more primary citations for introduction to replace reviews
- AAC(3)-XI citation (line 54) incorrectly placed, it is a member of the GNAT superfamily

Remaining concerns:

(2) Reply from first review revisited: When testing the functionality of the new AMEs we screened for resistance against 8 different aminoglycosides primarily selected based on clinical relevance. We agree with the reviewer that it would be of interest to examine the complete resistance profile of the newly discovered AMEs, however, due to the large number of genes, complete characterization is not feasible to do within a single study. In addition to testing the susceptibility against more compounds, we also argue that the genes need to be assessed in more hosts than *E. coli* to properly assess their clinical impact. Since both these aspects require significant undertakings, we see them as beyond the scope.

(2a) I agree that obtaining a complete resistance profile would require significant undertakings for this study and is beyond the scope. Access to certain compounds can also be a limitation. As stated by the authors, emerging ARGs are often identified after they are widespread in pathogens (line 69). Strategies moving forward in surveillance, countermeasures etc., thus need to include therapeutics currently in development so that they are not met with resistance immediately after approval or soon after. Within the panel of 7 aminoglycosides tested, there exists enough structural similarities that at least one of the compounds could have been substituted for a distinct scaffold that is also currently in clinical development. This would have increased the significance for their models' ability to identify emerging threats to aminoglycosides currently in use as well as inform stewardship programs for next-generation aminoglycosides in

development.

(2b) Furthermore, I agree these genes would need to be tested in additional hosts to properly assess their clinical impact. As discussed in responses no.4 & no.5 to the other reviewer, the authors note their use of the microbiological definition of resistance and only highlight the levels of resistance above clinical breakpoints as it relates to the potential of a gene. This is supported by the authors investigation aim to predict genes increasing resistance above that of wild-type bacteria. That being said, the authors are proposing that the prediction capabilities of their models can contribute to protecting antibiotics. Their conclusions section also stress the importance of implementing suitable preventive measures limiting dissemination. This should then include protecting aminoglycosides in development – as they are also clinically relevant – so that emerging or potential ARGs are identified prior to becoming widely disseminated.

(3) I can understand why AAC(3)-IV was not included in Model C (assuming distance from other AACs in the tree?), however, there was no mention of this in the manuscript. It would be helpful to the readers to provide reasons for its exclusion and thus limitation of their models predicting resistance. AAC(3)-IV provides broad spectrum aminoglycoside resistance, it's unfortunate that it was not included.

(4) I agree with the other reviewer that AAC(6')-Ib would have been a more appropriate positive control. This enzyme confers a significant level of resistance to amikacin and considering the unexpected lack of resistance observed with APH(3')-III should have been included. In addition to this point, I noticed D292 confers resistance to amikacin and not gentamicin yet Supplementary Table 2 shows a closer sequence similarity to AAC(6')-IIb, which does not confer amikacin resistance and should confer gentamicin resistance. An appropriate control for amikacin under these experimental conditions would provide confidence in this phenotype. Or maybe this was a typo in the table?

Reviewer #2 (Remarks to the Author):

The authors have put in a lot of effort in addressing my previous comments and have made the necessary edits, plus a few additional updates.

The new version of the manuscripts is clearly improved by providing an increased level of clarity and explanations regarding the interpretations.

I have no further comments regarding the manuscript.

Response to review comments.

Reviewer 1

Summary: In their revised manuscript, the authors made significant improvements to the text which presents a much clearer narrative. Following the suggestions from both reviewers, the authors clearly define terms and provide additional details for their analyses to prevent over-interpretation of the data. The authors end their manuscript with a thought-provoking discussion and provide directions for future studies.

Minor notes

1. *Comment:* Appropriate corrections have been made to the phylogenies and nomenclature adjustments for model references prevent confusion. These changes add to the manuscript's potential impact, improving surveillance for a class of modifying enzymes often faced with annotation inconsistencies in public databases. Great improvements were made on the presentation of phylogenies in supplementary data figures, easier to read.
 - Minor suggestion would be to find more primary citations for introduction to replace reviews
 - AAC(3)-XI citation (line 54) incorrectly placed, it is a member of the GNAT superfamily

Reply: We thank the reviewer for the continued constructive feedback. We have included additional references in the introduction, citing primary research papers. The AAC(3)-XI citation on line 54 was indeed misplaced, and has now been removed.

2. *Comment:* Reply from first review revisited: When testing the functionality of the new AMEs we screened for resistance against 8 different aminoglycosides primarily selected based on clinical relevance. We agree with the reviewer that it would be of interest to examine the complete resistance profile of the newly discovered AMEs, however, due to the large number of genes, complete characterization is not feasible to do within a single study. In addition to testing the susceptibility against more compounds, we also argue that the genes need to be assessed in more hosts than *E. coli* to properly assess their clinical impact. Since both these aspects require significant undertakings, we see them as beyond the scope.
 - a. I agree that obtaining a complete resistance profile would require significant undertakings for this study and is beyond the scope. Access to certain compounds can also be a limitation. As stated by the authors, emerging ARGs are often identified after they are widespread in pathogens (line 69). Strategies moving forward in surveillance, countermeasures etc., thus need to include therapeutics currently in development so that they are not met with resistance immediately after approval or soon after. Within the panel of 7 aminoglycosides tested, there exists enough structural similarities that at least one of the compounds could have been substituted for a distinct scaffold that is also currently in clinical development. This would have increased the significance for their models' ability to identify emerging threats to aminoglycosides currently in

use as well as inform stewardship programs for next-generation aminoglycosides in development.

- b. Furthermore, I agree these genes would need to be tested in additional hosts to properly assess their clinical impact. As discussed in responses no.4 & no.5 to the other reviewer, the authors note their use of the microbiological definition of resistance and only highlight the levels of resistance above clinical breakpoints as it relates to the potential of a gene. This is supported by the authors investigation aim to predict genes increasing resistance above that of wild-type bacteria. That being said, the authors are proposing that the prediction capabilities of their models can contribute to protecting antibiotics. Their conclusions section also stress the importance of implementing suitable preventive measures limiting dissemination. This should then include protecting aminoglycosides in development – as they are also clinically relevant – so that emerging or potential ARGs are identified prior to becoming widely disseminated.

Reply: We fully agree with the reviewer that it would be of interest to evaluate if the new genes can provide resistance against aminoglycosides currently in development. We will take these recommendations into account when designing future research projects.

3. *Comment:* I can understand why AAC(3)-IV was not included in Model C (assuming distance from other AACs in the tree?), however, there was no mention of this in the manuscript. It would be helpful to the readers to provide reasons for its exclusion and thus limitation of their models predicting resistance. AAC(3)-IV provides broad spectrum aminoglycoside resistance, it's unfortunate that it was not included.

Reply: The phylogenetic distance to AAC(3)-IV was indeed the reason that it could not be included in model C while retaining sufficient performance of the model. The same was true for some other resistance determinants which our models were unable to capture. A sentence about this has been added to the methods section (Line 341-343).

4. *Comment:* I agree with the other reviewer that AAC(6')-Ib would have been a more appropriate positive control. This enzyme confers a significant level of resistance to amikacin and considering the unexpected lack of resistance observed with APH(3')-III should have been included.

In addition to this point, I noticed D292 confers resistance to amikacin and not gentamicin yet Supplementary Table 2 shows a closer sequence similarity to AAC(6')-IIb, which does not confer amikacin resistance and should confer gentamicin resistance. An appropriate control for amikacin under these experimental conditions would provide confidence in this phenotype. Or maybe this was a typo in the table?

Reply: We appreciate the reviewer's concern. There was no typo in the table. The sequence identity between D292 and AAC(6')-IIb was very low (43.4% amino acid level), so it is reasonable that the two proteins would not be functionally identical. Indeed, D292 clearly did not provide resistance against gentamicin (where appropriate controls were in place) which supports the notion that it does not confer an AAC(6')-II phenotype. As we mentioned in our previous reply to Reviewer 2, while it is unfortunate that our positive control for amikacin was not functional, we see no reason to doubt our experimental results given that all other controls behaved as expected.